# Differential Effect of Simulated Microgravity on the Cellular Uptake of Small Molecules

**DOI:** 10.3390/pharmaceutics16091211

**Published:** 2024-09-14

**Authors:** Odelia Tepper-Shimshon, Nino Tetro, Roa’a Hamed, Natalia Erenburg, Emmanuelle Merquiol, Gourab Dey, Agam Haim, Tali Dee, Noa Duvdevani, Talin Kevorkian, Galia Blum, Eylon Yavin, Sara Eyal

**Affiliations:** 1Institute for Drug Research, School of Pharmacy, Faculty of Medicine, The Hebrew University of Jerusalem, Jerusalem 9112002, Israel; odelis1@gmail.com (O.T.-S.); ntetro@gmail.com (N.T.); roua.hamed@mail.huji.ac.il (R.H.); natalia.erenburg@mail.huji.ac.il (N.E.); emmanuellem@savion.huji.ac.il (E.M.); gourab.dey@mail.huji.ac.il (G.D.); galiabl@ekmd.huji.ac.il (G.B.); eylony@ekmd.huji.ac.il (E.Y.); 2School of Space Science, Hebrew University Youth Division, The Hebrew University of Jerusalem, Jerusalem 9112002, Israelnoaduv123@gmail.com (N.D.);

**Keywords:** microgravity, multidrug resistance–associated proteins, MRP1, P-glycoprotein, MDR1, breast cancer resistance protein, BCRP, GB123, glucose transporters, pharmacokinetics

## Abstract

The space environment can affect the function of all physiological systems, including the properties of cell membranes. Our goal in this study was to explore the effect of simulated microgravity (SMG) on the cellular uptake of small molecules based on reported microgravity-induced changes in membrane properties. SMG was applied to cultured cells using a random-positioning machine for up to three hours. We assessed the cellular accumulation of compounds representing substrates of uptake and efflux transporters, and of compounds not shown to be transported by membrane carriers. Exposure to SMG led to an increase of up to 60% (*p* < 0.01) in the cellular uptake of efflux transporter substrates, whereas a glucose transporter substrate showed a decrease of 20% (*p* < 0.05). The uptake of the cathepsin activity-based probe GB123 (MW, 1198 g/mol) was also enhanced (1.3-fold, *p* < 0.05). Cellular emission of molecules larger than ~3000 g/mol was reduced by up to 50% in SMG (*p* < 0.05). Our findings suggest that short-term exposure to SMG could differentially affect drug distribution across membranes. Longer exposure to microgravity, e.g., during spaceflight, may have distinct effects on the cellular uptake of small molecules.

## 1. Introduction

Microgravity, defined as the condition of experiencing very low or negligible gravitational forces, is a crucial aspect of the space environment. The microgravity of space can considerably alter human physiology, affecting almost every system of the body. Examples include fluid shifts toward the head and chest, an approximately 20-fold faster increase in carotid artery stiffness as compared to healthy aging, and accelerated bone and muscle mass loss [1].

The physiological changes that occur during spaceflight can result in medical conditions that require drug treatment and can affect the pharmacokinetics and pharmacodynamics of medications taken in space [1]. One factor that may contribute to altered pharmacokinetics and pharmacodynamics is an increase in membrane fluidity, which was observed during a parabolic flight mission and in simulated microgravity (SMG). In a study conducted onboard a sounding rocket, altered fluidity was shown to affect lidocaine integration into membranes. Microgravity was also associated with reduced transmembranal transport activity of the multidrug resistance–associated transporter (MRP) 2 [1].

Based on the changes in membrane properties, we hypothesized that microgravity could affect the distribution of small molecules into cells, due to changes in passive diffusion, carrier-mediated transport, or both. In this pilot study, we tested our hypothesis using SMG. We focused on the activity of the efflux transporters P-glycoprotein (P-gp), the breast cancer resistance protein (BCRP), and MRP1. We used selective transporter inhibitors and cell lines that overexpress these carriers to isolate the effects of SMG on distinct transporters. We also assessed the distribution into cells of an uptake transporter substrate, a compound which is not a P-gp/BCRP substrate, and several compounds of larger molecular weights. Our preferred probes were molecules trapped within cells (Table 1), which helped combat the reversibility of SMG effects when cells are removed from the microgravity simulator.

## 2. Materials and Methods

### 2.1. Materials

GB123 [2] and the FITC-labeled nuclear localization sequence (NLS)-peptide nucleic acid (PNA) [3] were synthesized as previously described. Bovine serum albumin (BSA), Biotechnology Grade, was obtained from Tamar laboratory supplies (Mevaseret Zion, Israel). The other cell culture reagents, phosphate-buffered saline (PBS), trypsin EDTA Solution C (0.05%), and 2,3-Bis(2-methoxy-4-nitro-5-sulfophenyl)-2H-tetrazolium-5-carboxanilide (XTT) were purchased from Biological Industries Ltd. (Beit Haemek, Israel). Dimethyl sulfoxide (extra dry 99.7% DMSO) was obtained from Acros Organics (Geel, Belgium). Calcein AM and boron-dipyrromethene (BODIPY) prazosin were obtained from Thermo Fisher Scientific (Waltham, MA USA). Hoechst 33342 was obtained from Abcam (Cambridge, UK). 2-Deoxy-2-[(7-nitro-2,1,3-benzoxadiazol-4-yl)amino]-D-glucose (2-NBDG) was obtained from Cayman Chemical (Ann Arbor, MI, USA). 4′,6-Diamidino-2-phenylindole (DAPI) was obtained from Fluoromount G (SouthernBiotech, Birmingham, AL, USA). Valspodar (PSC-833) was obtained from Tocris Bioscience (Bristol, UK). All other reagents were obtained from Sigma-Aldrich (Rehovot, Israel).

### 2.2. Cell Lines and Cell Culture

The cell lines were selected based on their profile of efflux transporter expression. Several species were chosen in order to enhance extrapolation potential. The RAW 264.7 line is derived from murine macrophages and has been commonly utilized in microgravity research [4]. These cells express functional P-gp [5]. Madin–Darby canine kidney (MDCK) II cells transfected with cDNA coding for the human P-gp (MDCK-MDR1 cells) are a well-characterized in vitro model for assessment of P-gp activity [6,7,8]. U87 human glioblastoma cells were selected based on their low level of P-gp expression [9].

RAW 264.7 cells (from Prof. Boaz Tirosh, The Hebrew University) were grown in RPMI 1640 medium supplemented with 10% fetal bovine serum, L-glutamine (2 mM), sodium pyruvate (1 mM), non-essential amino acids, penicillin (100 Units/mL), and streptomycin (100 μg/mL). U87-MG cells (from Prof. Ofra Benny, The Hebrew University) were cultured in Eagle’s MEM medium supplemented with 10% fetal calf serum, L-glutamine (2 mM), sodium pyruvate (1 mM), and 1% penicillin/streptomycin. MDCK-MDR1 cells were provided by Prof. Alfred Schinkel (The Netherlands Cancer Institute). Cells were maintained in Dulbecco’s modified Eagle’s phenol-free low-glucose medium (DMEM) supplemented with 10% fetal bovine serum, 2 mM L-glutamine, 100 units/mL penicillin, and 100 μg/mL streptomycin at 37 °C in a 5% CO_2_ incubator. The cells were harvested after achieving 70–80% confluence. The transmembranal transfer studies were all conducted using suspended cells. Cell viability was analyzed using the XTT assay kit according to the manufacturer’s instructions.

### 2.3. Tested Compounds

Calcein AM and Hoechst 33342 were used at 0.25 μM [10,11] and 60 μM (determined after several preliminary experiments), respectively. Doxorubicin was used at 5 μM, based on preliminary XTT experiments showing that at this concentration, it does not affect the viability of the studied cells. The concentration of BODIPY prazosin was 1 μM (based on preliminary experiments) and that of 2-NBDG was 100 μM [12]. Valspodar (a P-gp inhibitor [13]) was used at 1.65 μM [14] and fumitremorgin C (FTC; a BCRP inhibitor [13]) at 10 μM [14,15]. FITC-labeled dextrans were used at 2 mg/mL [16,17] (molar values are only an approximation, see Table 1) and NLS-PNA-FITC was used at 5 μM [3].

### 2.4. Experiments in Simulated Microgravity

Microgravity conditions were simulated by a random positioning machine (RPM) 2.0 (purchased from Yuri GmbH, Meckenbeuren, Germany; originally from DutchSpace Airbus, Leiden, The Netherlands). The RPM continuously and randomly changes the orientation of the accommodated experiments relative to the Earth gravity vector. Objects cannot adjust or react to the gravitational pull due to rapid and continuous changes in orientation. This simulates a microgravity environment by neutralizing the net effect of gravity on the objects, allowing gravity levels that replicate those of the Moon (0.16 g) and of Mars (0.38 g), with a minimum of 10^−3^ g. The RPM was maintained within a cell culture incubator (3 °C, 5% CO_2_).

Cells were removed from the culture plates according to the instructions from the American Type Culture Collection (ATCC) website; 0.05% trypsin for MDCK cells, 0.25% trypsin for U87 cells, and scraping for RAW 264.7 cells. Cells in media containing the fluorescent probe were transferred to 0.5 mL Eppendorf tubes placed at the center of the simulator (Appendix A, Figure A1) or on a shelf within the same incubator. Potential effects of the stationary position were excluded based on comparison to samples placed on a shaker (Appendix B, Figure A2). To enable the placement of all tubes on the central part of the RPM, all experiments were conducted using up to six tubes per treatment group. Incubation in the RPM or under control conditions was for the duration of one hour unless otherwise stated (due to slow uptake kinetics). In experiments with transporter inhibitors, the cells were preincubated with the inhibitor for one hour under Earth conditions. The inhibitor was also present throughout the experiment. The RPM was set at Partial G, Loop path, and No Motion mode.

At the end of the experiments, all tubes were immediately placed on ice and the remaining procedures were carried out at 4 °C. The cells were washed with PBS, filtered using a 40 µm nylon mesh filter, and fluorescence was measured by flow cytometry using a Fortessa FACS Analyzer (BD LSRFortessa™ Flow Cytometer, BD Biosciences, San Jose, CA, USA). The results were analyzed using FlowJo™ v10.7 analysis software (https://www.flowjo.com; accessed on 1 August 2024).

For the analyses with GB123, RAW 264.7 cells were divided into four groups. Two groups were incubated with 0.25 μM GB123 for two hours, washed three times, and incubated with PBS for an additional two hours. The other two groups were incubated with the medium over the first two hours, then with 0.25 μM GB123 under Earth conditions for the next two hours. In the cellular uptake experiments, the chambers were washed with phosphate-buffered saline (PBS) three times and stained with DAPI (1:1000) diluted in PBS. The cells were then imaged by a Nikon motorized Ti2E confocal fluorescent microscope with a Yokogawa W1 Spinning Disk (Olympus, Tokyo, Japan). The cathepsin binding analysis was conducted after the cells were lysed in RIPA buffer. Equal amounts of protein (100 μg per sample) were loaded onto the gel and separated on a 12.5% SDS PAGE (Appendix D, Figure A3). Instead of transferring the proteins to a membrane and using a loading control, we scanned the gel for Cy5 using a Typhoon laser scanner and compared the fluorescence intensity of GB123 between the groups. This method allowed direct assessment and comparison of cathepsin expression levels [14].

#### Statistical Analysis

Two-way ANOVA, the Mann–Whitney test or the Kruskal–Wallis test, as appropriate, were used for comparisons across treatment groups (Prism 10.2.3; GraphPad, La Jolla, CA, USA). Flow cytometry results are expressed as the medians of the fluorescence signal obtained in 10,000 cells. Results are reported as individual values with group medians. A *p* value ≤ 0.05 was considered significant.

## 3. Results

### 3.1. Simulated Microgravity Can Enhance the Cellular Uptake of Efflux Transporter Substrates

Exposure to SMG increased the cellular uptake of calcein AM (Figure 1a–c). The effect was highest in U87 cells (a median 60% increase in emission; Figure 1b). Overall, the magnitude of change in calcein AM emission did not reflect P-gp abundance. However, in MDCK-MDR1 cells, the SMG-inhibition interaction was significant, implying that both efflux transport and diffusion were affected (Figure 1c). In the doxorubicin experiment, only the effect of gravity was statistically significant (Figure 1d). The fold change in doxorubicin (Figure 1d) and BODIPY prazosin uptake (Figure 1e) was similar to that of calcein AM. Hoechst 33342 emission was not significantly affected by SMG (Figure 1f). We did not pursue the experiments with BODIPY prazosin in MDCK-MDR1 and U87 cells any further because these cells have no functional BCRP activity [18] or express low BCRP levels [9].

### 3.2. Glucose Transporter Activity/Metabolism Is Reduced in Simulated Microgravity

Unlike the efflux transporter substrates, the emission intensity of the glucose analog 2-NBDG was 20% lower under SMG conditions targeting 0.001 g (*p* < 0.01; Appendix E, Figure A4). Similar results were obtained at 0.16 g.

### 3.3. Simulated Microgravity Can Enhance the Cellular Uptake of a Molecule Larger Than 1000 g/mol

GB123, a cathepsin activity-based probe, is larger than the transporter substrates described above (Table 1). Under simulated Moon gravity (0.16 g), the cellular accumulation of this compound was also enhanced by 30% (Appendix F, Figure A5a–c). The experiments were conducted at higher gravity as compared to the others in order to minimize the potential of cathepsin activation. Indeed, the change in GB123 uptake was not due to increased cathepsin activity (which tended to decrease and not increase under SMG conditions; Appendix F, Figure A5d,e).

### 3.4. The Cellular Distribution of Molecules that Weigh 3000 g/mol or More Is Reduced in Simulated Microgravity

To assess the effect of size and charge on cellular uptake in SMG, we used positively charged, negatively charged, and neutral FITC-labeled dextrans. The signal from neutral dextran was similar to that of non-treated cells and we did not further explore this compound. Positive dextran yielded a considerably higher signal than negative dextran. The emission of both dextrans was approximately half in SMG (Figure 2a,b).

The effect of SMG on large, positively charged molecules was confirmed with NLS-PNA-FITC, [6253.70 g/mol] whose fluorescence intensity at SMG was lower as well (Figure 2c,d).

## 4. Discussion

Since the beginning of human spaceflight, significant time, resources, and effort have been devoted to the study of space physiology, whereas space pharmacology was far less thoroughly explored. Few pharmacokinetic studies have been conducted in astronauts and cosmonauts, with conflicting findings. The relatively consistent delayed absorption and other pharmacokinetic changes were mostly attributed to fluid shifts and changes in gastrointestinal function [1]. However, the microgravity of space might also affect membrane permeability, as was previously demonstrated in isolated vesicles [19]. This prompted us to explore the transmembranal distribution of smaller and larger molecules. We investigated small molecule (<1000 g/mol) efflux transporter substrates, an uptake transporter substrate, and larger molecules to rule out globally enhanced diffusion into cells. To avoid altered gene expression as a confounder, we limited our analyses to one hour (except for the GB123 and NLS-PNA-FITC studies).

This analysis is a pilot study, yet we could identify several physicochemical properties of the tested compounds that could affect their behavior in SMG. To our knowledge, such an analysis has not yet been reported. One notable observation was the higher cellular uptake of efflux transporter substrates in SMG. The magnitude of change was modest but statistically significant. The SMG effect on calcein AM kinetics was independent of P-gp abundance (Figure 1a–c; low, moderate, and high in U87 [9], RAW 264.7 [5], and MDCK-MDR1 cells, respectively). It was likely related to both enhanced diffusion and reduced P-gp activity (Figure 1c). The altered P-gp function could reflect changes in substrate or ATP binding, altered internalization, or cytoskeletal changes. The reduced 2-NBDG accumulation coupled with the lack of SMG effect on Hoechst 33342 suggest that the higher distribution of small molecules into cells in SMG is not a global phenomenon.

The SMG effect on a larger molecule, GB123, was similar to those observed with the efflux transporter substrates (although GB123 is not a P-gp or BCRP substrate [14]). Our analysis excluded enhanced cathepsin activity as the cause because activity tended to be lower when the cells were exposed to SMG. The enhanced GB123 uptake did not extend to larger molecules, at least not to those with the molecular properties described above.

This pilot study has several limitations. First, lower gravity (0.001 g) is achieved only after prolonged incubation (e.g., 24 h) and the one-hour exposure did not suffice. Yet, gravity was 0.01 g at 10 min and 0.006 g at 30 min after the onset of the experiment (Appendix C). This issue was less pronounced with lunar gravity, because the target value (0.16 g) was obtained within 30 min. Second, the closed Eppendorf tubes minimized exposure to CO_2_, which could potentially affect the results However, preliminary experiments with Falcon flasks that allow gas exchange yielded similar results, probably because the incubation was relatively short. Third, we took all available steps in order to remove bubbles, but their presence in the vials (and consequent shear forces) cannot be completely ruled out. Therefore, our findings might misinterpret the magnitude of the SMG effect. Additionally, this study assessed only short-time exposure and only transcellular transport. Longer exposure to microgravity has been shown to affect transporter gene expression [1] and could change the magnitude or reverse the direction of the changes observed in the current study. Altered paracellular transport (e.g., tight junction opening) might potentially also affect the transfer of small and large molecules across biological barriers in space. Finally, the cells were cultured in 2D. A study in cardiomyocytes demonstrated similar cellular responses to pharmacological agents in 2D and 3D under SMG conditions [20]. Hence dimensionality is not necessarily expected to affect our results, but further experiments should be performed to ensure that this is the case.

If our findings are applied to humans, membrane permeability to small- and large-molecule drugs could be altered. This could result in enhanced absorption, reduced systemic elimination, and enhanced distribution across the BBB, which may require dosing adjustment. One example is apixaban, a small molecule, P-gp substrate oral anticoagulant [21].

## 5. Conclusions

SMG selectively affects the uptake of small molecules into cells, but the direction of change varies across molecules. The changes were observed across several cell lines of various origins, suggesting that this phenomenon is not cell-type specific. In space, this could alter drug absorption, distribution, and elimination, and potentially also drug effects. Future research may shed light on the clinical implications of altered transmembranal transport of small molecules during spaceflight.

## Figures and Tables

**Figure 1 pharmaceutics-16-01211-f001:**
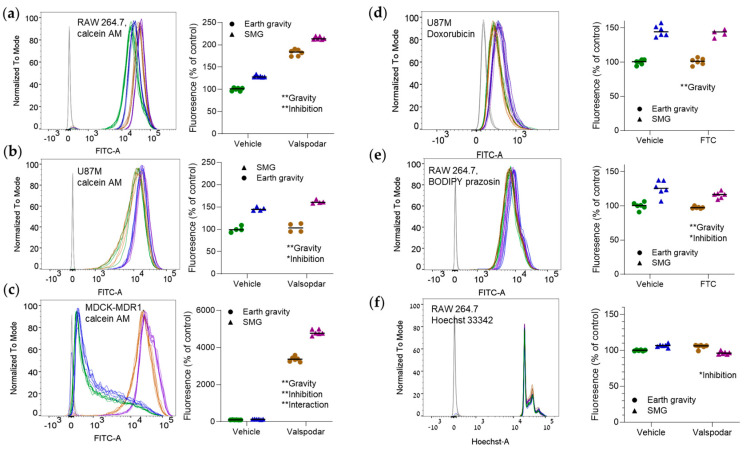
The uptake of efflux transporter substrates can be increased in simulated microgravity (SMG). Cells were incubated at Earth gravity or SMG (target: 0.001 g; Appendix C) with efflux transporter substrates in the presence or the absence of transporter inhibitors. Emission intensity was measured by flow cytometry. The left part of each panel shows a representative emission distribution curve. The right part is a quantitative analysis of the results. Shifts to the right indicate a higher uptake of the fluorescent probe by the cells. (**a**–**c**) Raw 264.7 (n = 6/group) (**a**), U87 (n = 4/group) (**b**), or MDCK-MDR1 (n = 6/group) (**c**) cells, incubated with 0.25 μM calcein AM with or without 1.65 μM valspodar (a P-gp inhibitor) [13]). (**d**) U877 cells incubated with 5 μM doxorubicin with or without 1.65 μM valspodar (n = 6/group). (**e**) RAW 264.7 cells incubated with 1 μM BODIPY prazosin with or without 10 μM fumitremorgin C (FTC; a BCRP inhibitor [13]; n = 6/group). (**f**) RAW 264.7 cells incubated with 60 μM Hoechst 33342 with or without 1.65 μM valspodar (n = 6/group, except for n = 5 in the SMG + valspodar group). Dark gray lines in the distribution curves denote non-stained cells. * *p* < 0.05; ** *p* < 0.01; 2-way ANOVA. The test assesses whether the emission is significantly affected by SMG, the inhibitor, and their interaction. The figures are labeled accordingly. Experiments were repeated thrice with similar results. The numbers of replicates per group denote biological replicates (samples cultured separately). Heights are normalized (“to mode”) to highlight shifts to the right or the left. Green, vehicle and Earth gravity; blue, vehicle and SMG; brown, transporter inhibitor and Earth gravity; purple, transporter inhibitor and SMG.

**Figure 2 pharmaceutics-16-01211-f002:**
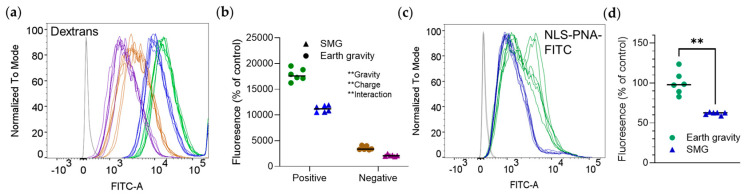
Simulated Moon gravity reduces the uptake of molecules larger than 3000 g/mol into RAW 264.7 cells. Cells were incubated at Earth gravity or SMG with positively charged, negatively charged, or neutral dextrans (Table 1) for one hour, or with NLS−PNA−FITC for three hours. Emission intensity was measured by flow cytometry. Shown are representative distribution curves of emission intensity obtained by flow cytometry (**a**,**c**) and quantitative analyses of the results (**b**,**d**). (**a**,**b**) Fluorescently labeled dextrans. (**b**,**c**) NLS−PNA−FITC. ** *p* < 0.01, two−way ANOVA (**b**) or Mann-–Whitney test (**d**). Experiments were repeated thrice (n = 6 biological replicates per group in each experiment), with similar results. (**a,b**) Green, positively-charged dextran and Earth gravity; blue, positively-charged dextran and SMG; brown, negatively-charged dextran and Earth gravity; purple, negatively-charged and SMG.

**Table 1 pharmaceutics-16-01211-t001:** Physicochemical properties of the studied compounds.^1^

Compound	MW (g/mol)	xLogP	HBD	HBA	TPSA	Comments
2-NBDG	342	−1.4	5	11	195	A fluorescently labeled deoxyglucose analog; a probe of glucose carrier activity that becomes trapped in cells (hence results should not be affected by SMG reversal)
Hoechst 33342	453	4.6	2	5	73	A DNA fluorescent stain; a P-gp and BCRP substrate
Doxorubicin	543	1.3	6	12	206	A fluorescent chemotherapeutic agent that intercalates into DNA and inhibits topoisomerase II; a substrate of P-gp, BCRP, and MRP1
BODIPY prazosin	563	N/A	1	10	111	A fluorescent probe of BCRP activity
Calcein AM	995	2.9	0	25	305	A non-fluorescent precursor of calcein
Calcein	623	−3.1	6	15	232	A fluorescent probe of P-gp activity; also an MRP1 substrate
GB123	1198	N/A	N/A	N/A	N/A	A fluorescently quenched activity-based probe of cysteine cathepsin activity; not a P-gp or BCRP substrate
NLS-PNA-FITC	6254					An 18-mer peptide nucleic acid (PNA) conjugated to FITC fluorophore and to the NLS peptide, which was previously reported as a nuclear targeting agent

^1^ Data for compounds other than GB123 and NLS-PNA-FITC are based on PubChem (https://pubchem.ncbi.nlm.nih.gov/; accessed on 26 August 2024). The molecular weights of dextrans were 4000 g/mol (neutral), 3000–6000 g/mol (positively charged), and 4000 g/mol (negatively charged). BCRP, breast cancer resistance protein; Calcein AM, calcein acetoxymethyl ester; FITC, fluorescein isothiocyanate; HBA, hydrogen bond acceptor count; HBD, hydrogen bond donor count; MRP, multidrug resistance–associated protein; N/A, not available; 2-NBDG, 2-Deoxy-2-[(7-nitro-2,1,3-benzoxadiazol-4-yl)amino]-D-glucose; NLS, nuclear localization sequence; P-gp, P-glycoprotein; xLogP; calculated logarithm of the predicted n-octanol: water partition coefficient; TPSA, topological polar surface area. References for the table are available upon request.

## Data Availability

Data are available from S.E. upon request.

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
