# Peer review of "Differential Effect of Simulated Microgravity on the Cellular Uptake of Small Molecules"

_pharmaceutics, 2024, doi:10.3390/pharmaceutics16091211_

Round 1

Reviewer 1 Report

Comments and Suggestions for Authors

The authors explored the potential utility of short-term exposure to microgravity for enhancing the cellular uptake of therapeutic compounds, based on reported microgravity-induced changes in membrane properties.

The findings are of interest but there were a number of issues which should be addressed to  improve this manuscript.

Under Introduction

1)    First paragraph: A number of issues including lack of clear flow, the paragraph jumps from one idea to another without clear transitions or explanations. It doesn't explain how the impact of microgravity on organs leads to the need for drug treatments and how this relates to the view of space as akin to aging and disease. In addition, phrases are repeated or rephrased in a way that doesn't add clarity.

2)    Line 36-37 should provide more context regarding statement of “one remarkable finding” what system, under what conditions, etc.

3)    Second paragraph also has issues, it is unclear what "Our focus" is referring to—it's whether it refers to the current study or past research, in addition "To account for rapid reversal" is unclear and lacks context. Lastly, the paragraph jumps between ideas without smooth transitions.

Under Materials and Methods

Section 2.2

1)    The rationale for the choice of cell lines should be included. Full description of the tissue of origin for each should also be noted and rationale for choice of each.

2)    It would be helpful to have a brief description about each chemical in the table and what is known about it in relation to studying membrane transport. For example, Doxorubicin: A chemotherapeutic agent known for its ability to intercalate DNA and inhibit topo II. It’s also fluorescent, making it useful for studying drug uptake and efflux mechanisms.

Section 2.4 Microgravity

3)    Experiments should note the settings for the RPM. What was the calculated gravity for the machine when used for each application?

4)    Why was trypsin used for two cell lines and scraping for the RAW 264.7 cells to detach from their substrate?

5)    Please note exact number of tubes for each treatment group (it is not correct to just say “up to 6 tubes/treatment group” this can be anywhere from 1-6)

6)    How was it ensured in capping the Eppendorf that no small bubbles formed under the lid which could then result in shearing forces under the sim microgravity. Typically, it is important that the containers used are filled in such a manner that no bubbles can be introduced. For example injecting to completely fill.

7)    Would be useful to have a small image of the placement of the samples on the microgravity simulator in appendix.

8)    Pore size of the nylon filter used should be mentioned.

9)    What is meant by the statement “culturing was for one hour” is this in reference to the time it took to trypsinize and set up each cell line in the tubes with the reagent? Why is this important? Did some tubes sit longer prior to initiating the simulated microgravity, so that some were sitting much longer under non-microgravity conditions than others? This should be clarified.

10) Lines 102-104, would be clearer to state: “Objects are unable to adjust or react to the gravitational pull due to rapid and continuous changes in orientation. This simulates a microgravity environment by neutralizing the net effect of gravity on the objects.”

Under Results section

Section 3.1

11) Line 142, missing a word and extra words not needed: “increase the cellular uptake of calcein AM in all the cell lines that were evaluated….

12) Line 146, Instead of stating “emission was marginal (Figure 1f) should state if it was significant or not.

13) If there are only two points being compared in the graphs, technically one cannot say "gravity level"  for example: “accumulation was affected by the gravity level”. This would imply multiple gravity levels were tested. Better to state effected by simulated microgravity or moon gravity exposure.

14) There is no rationale for why LPS was added, should be noted.

15)  Please indicate in the methods exact number of technical Replicates or multiple measurements within each biological replicate (e.g., triplicate measurements per sample to account for technical variability), biological replicates (parallel measurements of biologically distinct samples that capture biological variation. These replicates are treated identically within a single experimental run) and independent experiments (samples in which you repeat the entire study, often on different days or under slightly different conditions. This ensures that the results are not specific to a single experimental setup and are reproducible across different contexts) were done. It is noted on line 169 “Experiments were repeated at least twice with similar results”, does this mean it is a biological replicate or an independent experiment where different cells were thawed and cultured for these experiments on different days? Typically 3 independent experiments should be performed.

16) Figure 1 is very confusing; it would be better to have a different symbol for simulated microgravity rather than a star. It would be clearer just to have a legend with Earth gravity, and sim µg and different symbol for each, rather than the 1.0 and 0.001. The star for the sim µg is confusing given it looks similar to an asterisk.  It is not clear what the “**Gravity, **LPS” etc. mean. In the legend it only notes *p<0.05, so if this is a level of significance, what significance is ** ? Also what is being compared for defining this level of significance, should be shown. Again, not correct to state “gravity level” it is just sim µg or earth gravity, etc. What is the “gravity level” referring to in the figure? It is placed under "No inhibitor" and "with inhibitor" making the figure even more confusing. The font for the X and Y axis headings for the FACS profiles should be the same size as the other figures, and it should be noted that black peak is cells without component (if this is the case). Why do (d) and (e) not have cells without reagent shown in the FACS?

17) Rationale for not showing U87M and MDCK-MDR1 lines with Bodipy prazosin and Hoechst should also be stated.

18) Should state “simulated microgravity” not microgravity throughout the manuscript, it can be abbreviated once it is introduced as “sim µg” to save space.

19) Should note in figure 1 legend what valspodar does in parenthesis.

Figure 2

20) Similarly, would need to increase X and Y axis for FACS profile and include non-stained cell peak.

21) Please include in appendix the FACS gating strategy, showing gating of cells and what was pulled into the final profile shown.

22) For (b) and (c) please remove “gravity level” and make changes suggested for Fig. 1.

23) Should include error bars for figures.

24) Please note why the BeWo cell lines were chosen to investigate moon gravity 0.16, and why only for glucose transporter activity, not other lines and other compounds?

Section 3.3

25) Line 184, should note what P-gp and BCRP stand for and make clear why relevant.

Fig. 3

26) Why was GB123 only studied with moon gravity and in the RAW 264.7 cells? Please provide rationale.

27) Please note with an arrow which of the bands is being compared in the Western.

28)For graphs why is there an n of only 3 for (e)?

29) What was used as a loading control? There seems to be more loaded in last lane of group 2.

Section 3.4

       30) Line 201 -202: weight should be weigh.

       31) Fig. 4 again show X and Y labels on FACS panel in a bigger font, input the error bars for data, and change the symbols for earth and moon gravity and input legend with earth and moon rather than 1.0 and 0.16, so less confusing. Please again note in this and other figures what is being compared for significance by showing with comparison lines like in (d).

Under Discussion

        32) Line 234: “tree transporters…” should be three transporters.

        33) Please be clear in discussion when stating “the extent of this phenomenon was modest” whether it was significant or not.

        34)Line 252-53: better to state …”the cellular uptake of GB123 was higher under sim µg conditions.”

        35) Line 255: again not correct to state “in microgravity” better to state “lower under simulated microgravity conditions” both here and throughout the manuscript.

        36) Please check grammar through out there are a number of issues including lines 260-61 “ … the one-hr exposure did not allow achieving…”

        37) The last paragraph for the discussion is unclear, why was 1 hour chosen if the target gravity level was not able to be obtained? If it was 0.006 then why do the figures note 0.001? What levels were reached with the moon gravity in the experiments??

        38) It is noted that results are similar to when falcon flasks were used, however these would definitely have areas that could not be filled completely, causing bubbles that would be an issue when performing these studies.

        39)Line 269, should state more regarding what is meant and the significance of “Changes in para-cellular transport might matter as well” as it is written it is not very informative or clear.

        40) Similarly, what was noted in 3-D studies, how would you expect results to differ, what is the evidence for this?

Under Conclusions

      41)Line 277-278:  The authors can’t state that “Changes are observed across cell lines” as all cell lines were not investigated for all molecules.

       42) It is not clear how the authors envision the sim ug exposure to be used therapeutically, also what is the purpose of using moon gravity and why only with certain molecules?

Overall the manuscript would benefit from clearer explanations in several key areas: (1) Rationale for cell line selections, clarify why specific cell lines were chosen AND why they were used with particular molecules.  The criteria and reasoning should be clearly stated. (2) The choice of simulated µg vs moon gravity, the scientific rationale for this choice should be articulated clearly. (3) Therapeutic interventions, how would the authors envision their findings being applied therapeutically?

Comments on the Quality of English Language

The quality of English language needs some minor attention, mainly in some word choices, flow and structure. I have noted some issues in reviewer comments but the manuscript should be carefully reviewed in its entirety.

Author Response

Reviewer #1:

Under Introduction

  1. First paragraph: A number of issues including lack of clear flow, the paragraph jumps from one idea to another without clear transitions or explanations. It doesn't explain how the impact of microgravity on organs leads to the need for drug treatments and how this relates to the view of space as akin to aging and disease. In addition, phrases are repeated or rephrased in a way that doesn't add clarity.

The first paragraph was rewritten to improve flow and clarity (lines 31-36).

  1. Line 36-37 should provide more context regarding statement of “one remarkable finding” what system, under what conditions, etc.

Done (lines 45-51).

  1. Second paragraph also has issues, it is unclear what "Our focus" is referring to—it's whether it refers to the current study or past research, in addition "To account for rapid reversal" is unclear and lacks context. Lastly, the paragraph jumps between ideas without smooth transitions.

We revised the sentence and added a clarification, as follows: “Our preferred probes were molecules trapped within cells to combat the reversibility of SMG effects when cells are removed from the microgravity simulator (Table 1)” (lines 62-64).

Under Materials and Methods

Section 2.2

  1. The rationale for the choice of cell lines should be included. Full description of the tissue of origin for each should also be noted and rationale for choice of each.

Done (lines 94-100).

  1. It would be helpful to have a brief description about each chemical in the table and what is known about it in relation to studying membrane transport. For example, Doxorubicin: A chemotherapeutic agent known for its ability to intercalate DNA and inhibit topo II. It’s also fluorescent, making it useful for studying drug uptake and efflux mechanisms.

Done, and Table 1 was moved to the end of the introduction (P. 2).

Section 2.4 Microgravity

  1. Experiments should note the settings for the RPM. What was the calculated gravity for the machine when used for each application?

We added to the text (lines 159-160) the instrument settings. Appendix C now provides data on the actual gravity values obtained during the 0.001 g and 0.16 g studies. The experiments at 0.16 g were conducted over up to two hr, but the target gravity was achieved by 30 min.

  1. Why was trypsin used for two cell lines and scraping for the RAW 264.7 cells to detach from their substrate?

Cell removal from the culture plate was according to the instructions from ITCC website (lines 146-147).

  1. Please note exact number of tubes for each treatment group (it is not correct to just say “up to 6 tubes/treatment group” this can be anywhere from 1-6)

We added the number of replicates to the legend of each figure.

  1. How was it ensured in capping the Eppendorf that no small bubbles formed under the lid which could then result in shearing forces under the sim microgravity. Typically, it is important that the containers used are filled in such a manner that no bubbles can be introduced. For example injecting to completely fill.

We are aware of the bubble issue and have completed the filling with pipets. However, we cannot completely rule out the effects of bubbles. This was added to the paragraph discussing the study limitations (lines 354-357).

  1. Would be useful to have a small image of the placement of the samples on the microgravity simulator in appendix.

A photo was added as Appendix A (P. 11). The running letters of the other appendices were changed accordingly.

  1. Pore size of the nylon filter used should be mentioned.

The pore size was 40 µm. This was added to the methods section (lines 162-163).

  1. What is meant by the statement “culturing was for one hour” is this in reference to the time it took to trypsinize and set up each cell line in the tubes with the reagent? Why is this important? Did some tubes sit longer prior to initiating the simulated microgravity, so that some were sitting much longer under non-microgravity conditions than others? This should be clarified.

We added clarifications to this (lines 154-156).

  1. Lines 102-104, would be clearer to state: “Objects are unable to adjust or react to the gravitational pull due to rapid and continuous changes in orientation. This simulates a microgravity environment by neutralizing the net effect of gravity on the objects.”

Thank you for the suggestion, done (lines 140-142).

Under Results section

Section 3.1

  1. Line 142, missing a word and extra words not needed: “increase the cellular uptake of calcein AM in all the cell lines that were evaluated….

The sentence was revised (lines 192-194).

  1. Line 146, Instead of stating “emission was marginal (Figure 1f) should state if it was significant or not.

Done (lines 201-202).

  1.  If there are only two points being compared in the graphs, technically one cannot say "gravity level"  for example: “accumulation was affected by the gravity level”. This would imply multiple gravity levels were tested. Better to state effected by simulated microgravity or moon gravity exposure.

  1. Done, throughout the manuscript.

  1.  There is no rationale for why LPS was added, should be noted.

Given that the manuscript has become a Brief Report, with deleted the LPS studies to enhance clarity.

  1. Please indicate in the methods exact number of technical Replicates or multiple measurements within each biological replicate (e.g., triplicate measurements per sample to account for technical variability), biological replicates (parallel measurements of biologically distinct samples that capture biological variation. These replicates are treated identically within a single experimental run) and independent experiments (samples in which you repeat the entire study, often on different days or under slightly different conditions. This ensures that the results are not specific to a single experimental setup and are reproducible across different contexts) were done. It is noted on line 169 “Experiments were repeated at least twice with similar results”, does this mean it is a biological replicate or an independent experiment where different cells were thawed and cultured for these experiments on different days? Typically 3 independent experiments should be performed.

The majority of experiments were repeated trice. The experiments with GB123 were repeated twice All the indicated replicates are biological. We added this information to the legend of each figure.

  1.  Figure 1 is very confusing; it would be better to have a different symbol for simulated microgravity rather than a star. It would be clearer just to have a legend with Earth gravity, and sim µg and different symbol for each, rather than the 1.0 and 0.001. The star for the sim µg is confusing given it looks similar to an asterisk.  It is not clear what the “**Gravity, **LPS” etc. mean. In the legend it only notes *p<0.05, so if this is a level of significance, what significance is ** ? Also what is being compared for defining this level of significance, should be shown. Again, not correct to state “gravity level” it is just sim µg or earth gravity, etc. What is the “gravity level” referring to in the figure? It is placed under "No inhibitor" and "with inhibitor" making the figure even more confusing. The font for the X and Y axis headings for the FACS profiles should be the same size as the other figures, and it should be noted that black peak is cells without component (if this is the case). Why do (d) and (e) not have cells without reagent shown in the FACS?

Asterixis we substituted by triangles; we corrected the graph terminology; the font size of the FACS and Prisma results is now similar; We added an explanation for the statistical significance labeling and for the ** symbol; all experiments are now shown with the non-stained control. We slightly modified the figure to improve clarity (P. 6-7).

  1.  Rationale for not showing U87M and MDCK-MDR1 lines with Bodipy prazosin and Hoechst should also be stated

We added the explanation, as follows: “We did not further pursue the experiments with BODIPY prazosin in the MDCK-MDR1 and U87 cells as they have no functional BCRP activity [54] or express low BCRP levels [43]” (lines 207-210).

  1.  Should state “simulated microgravity” not microgravity throughout the manuscript, it can be abbreviated once it is introduced as “sim µg” to save space.

Wherever applicable, we corrected microgravity to simulated microgravity throughout the manuscript and the figures; we followed an NPJ Microgravity convention for simulated microgravity abbreviation – SMG.

  1. Should note in figure 1 legend what valspodar does in parenthesis.

The explanation was added to the legend, along with that for FTC (lines 222, 224).

Figure 2

  1. Similarly, would need to increase X and Y axis for FACS profile and include non-stained cell peak.

Because the manuscript type was changed to Brief Report, the figure became an Appendix (E; P. 13). The axes were modified.

  1.  Please include in appendix the FACS gating strategy, showing gating of cells and what was pulled into the final profile shown.

The gating strategy was added (Appendix E).

  1.  For (b) and (c) please remove “gravity level” and make changes suggested for Fig. 1.

Done.

  1.  Should include error bars for figures.

The figures show individual values that allow appreciation of the variation. We tried adding 95% CI but this reduced clarity.

  1.  Please note why the BeWo cell lines were chosen to investigate moon gravity 0.16, and why only for glucose transporter activity, not other lines and other compounds?

To standardize the experimental conditions, we repeated the 2-NBDG experiments with RAW 264.7 cells under target 0.001 g (Section 3.2. P. 7, and Appendix E). This was a pilot study. Within the deadline for the Special issue, we could not conduct further experiments with other uptake carrier substrates. Glucose transport was assessed because it's universal. 2-NBGD was selected because it becomes trapped within cells (Table 1).

Section 3.3

  1. Line 184, should note what P-gp and BCRP stand for and make clear why relevant.

P-gp and BCRP are now defined in the introduction (lines 57-58).

Fig. 3

  1. Why was GB123 only studied with moon gravity and in the RAW 264.7 cells? Please provide rationale.

An explanation was added, as follows: “This experiment was conducted at higher gravity as compared to the others, to minimize the potential of cathepsin activation” (lines 251-252).

  1.  Please note with an arrow which of the bands is being compared in the Western.

We added the information to the figure and its legend (Appendix F, P. 13).

  1. For graphs why is there an n of only 3 for e?

This analysis was conducted only to identify a potential cathepsin activation in SMG. Given the very low variability across triplicates, we did not further pursue this line of research.

  1.  What was used as a loading control? There seems to be more loaded in last lane of group 2.

The analysis was direct, as is now explained in the text: “Equal amounts of protein (100 μg per sample) were loaded onto the gel and separated on a 12.5% SDS PAGE. Instead of transferring the proteins to a membrane and using a loading control, we scanned the gel for Cy5 using a Typhoon laser scanner to compare the fluorescence intensity of GB123 between the groups. This method allowed direct assessment and comparison of cathepsin expression levels” (Lines 178-182). We added a figure of separated proteins as Appendix D (P. 12).

Section 3.4

  1. Line 201 -202: weight should be weigh.

Done (line 268).

  1. 4 again show X and Y labels on FACS panel in a bigger font, input the error bars for data, and change the symbols for earth and moon gravity and input legend with earth and moon rather than 1.0 and 0.16, so less confusing. Please again note in this and other figures what is being compared for significance by showing with comparison lines like in (d).

We corrected the figures as described for Figure 1. Unlike the Mann-Whitney test in (d), that compares two groups and the comparison can be shown as lines, the 2-Way ANOVA assesses factor effects, as now described in the legend of Fig. 1. Therefore, adding lines would be less appropriate.

Under Discussion

  1. line 234: “tree transporters…” should be three transporters.

The sentence was revised (lines 311-315).

  1. Please be clear in discussion when stating “the extent of this phenomenon was modest” whether it was significant or not.

Done (lines 312-313).

  1. Line 252-53: better to state …”the cellular uptake of GB123 was higher under sim µg conditions.”

The sentence was revised (lines 338-341).

  1. Line 255: again not correct to state “in microgravity” better to state “lower under simulated microgravity conditions” both here and throughout the manuscript.

Done, throughout the manuscript.

  1. Please check grammar through out there are a number of issues including lines 260-61 “ … the one-hr exposure did not allow achieving…”

Grammar was corrected by both applying Grammarly and by proof reading.

  1. The last paragraph for the discussion is unclear, why was 1 hour chosen if the target gravity level was not able to be obtained? If it was 0.006 then why do the figures note 0.001? What levels were reached with the moon gravity in the experiments??

This is indeed an inherent limitation of the work with the RPM – we tried to limit the exposure to minimize changes in gene expression but also wanted to allow sufficient time for some stabilization of gravity, as shown for 0.16 g. The relevant treatment groups are now titled “SMG” (and not 0.001 g). We added to Appendix C the data for both applications (see also response to Q. 3).

  1. It is noted that results are similar to when falcon flasks were used, however these would definitely have areas that could not be filled completely, causing bubbles that would be an issue when performing these studies.

We inspected carefully the tubes and did not identify large bubbles, but smaller ones cannot be excluded. We referred to the bubble limitation as described above (lines 354-357).

  1. Line 269, should state more regarding what is meant and the significance of “Changes in para-cellular transport might matter as well” as it is written it is not very informative or clear.

The sentence was modified to “Altered paracellular transport (e.g., tight junction opening) might also affect the transfer of small and large molecules across biological barriers in space [61]” (lines 361-363).

  1. Similarly, what was noted in 3-D studies, how would you expect results to differ, what is the evidence for this?

We modified the text, as follows: “Finally, the cells were cultured in 2-D. A study in cardiomyocytes demonstrated similar cellular responses to pharmacological agents in 2-D and 3-D under SMG conditions [62]. Hence dimensionality should not necessarily affect our results” (lines 363-366).

  1. Line 277-278: The authors can’t state that “Changes are observed across cell lines” as all cell lines were not investigated for all molecules.

We modified the sentence to “changes were observed across several cell lines” (line 381).

  1. It is not clear how the authors envision the sim ug exposure to be used therapeutically, also what is the purpose of using moon gravity and why only with certain molecules?

We added text related to potential clinical implications, as follows: “If our findings are translated to humans in space, membrane permeability to small- and large-molecule drugs could be altered in space. This could result in enhanced absorption, reduced systemic elimination, and enhanced distribution across the BBB in space which may require dosing adjustment. One example is apixaban [12,63], a small molecule, P-gp substrate oral anticoagulant [64]” (lines 369-373).

For space and clarity considerations, we deleted the therapeutic use on Earth from the introduction and the conclusions. See the response to Q. 27 regarding Moon gravity in the GB123 study.

Overall the manuscript would benefit from clearer explanations in several key areas: (1) Rationale for cell line selections (done, lines X-Y), clarify why specific cell lines were chosen AND why they were used with particular molecules.  The criteria and reasoning should be clearly stated. (2) The choice of simulated µg vs moon gravity, the scientific rationale for this choice should be articulated clearly. (3) Therapeutic interventions, how would the authors envision their findings being applied therapeutically?

Thank you for the detailed review. We added the requested explanations, as described above. We hope they are clear and sufficient.

Comments on the Quality of English Language

The quality of English language needs some minor attention, mainly in some word choices, flow and structure. I have noted some issues in reviewer comments but the manuscript should be carefully reviewed in its entirety.

Language and grammar were checked with Grammarly. The manuscript was also proofread.

Reviewer 2 Report

Comments and Suggestions for Authors

This manuscript entitled ‘Differential effect of simulated microgravity on the cellular uptake of small molecules’ by Tepper-Shimshon O. et al. studied the effect of microgravity on the cellular uptake of drugs/compounds. Fluorescence-based cellular uptake was studied under microgravity to evaluate the role of uptake and efflux transporters. The research topic is interesting, and a detailed exploration of the problem could have significantly contributed to the field. However, the manuscript only reports cellular uptake data without performing follow-up experiments to understand the actual mechanism. 

Experimental design issues, including the choice of cells and multi-species cell lines, are key concerns. Cell viability and the effect of microgravity on cellular architecture should have been evaluated. Both adherent and suspended cells were used in the study. For adherent cells, did the authors confirm that transporter activity was equivalent in suspension versus surface-adhered conditions? 

In section 3.3, cellular uptake was studied with only one molecule, and the confocal images have different backgrounds for the red color. These images need to be captured under higher power for accurate calculation, and a single molecular evaluation is not sufficient to draw conclusions about the class of drugs showing transporter-mediated uptake. Additionally, the Western blots in Figure 3d are not labeled for the proteins.

 Apart from experimental design issues, the manuscript requires significant improvement in writing. The Introduction and Results sections lack detail, and no information is provided on the basis of selection for drug molecules and cell lines, aside from some coverage in the Discussion. Overall, these findings appear preliminary and require further investigation for consideration for publication.

Author Response

The research topic is interesting, and a detailed exploration of the problem could have significantly contributed to the field. However, the manuscript only reports cellular uptake data without performing follow-up experiments to understand the actual mechanism.

In line with the Academic Editor's Notes, the manuscript format was converted into a Brief Report, implying that a detailed assessment of the mechanisms that affect transporter expression in microgravity would be beyond its scope.

Experimental design issues, including the choice of cells and multi-species cell lines, are key concerns.

The rationale for selecting the various cell lines was added to Section 2.2 (lines 94-100). Further details were added to the Methods section (Table 1 and throughout Section 2).

Cell viability and the effect of microgravity on cellular architecture should have been evaluated.

We added a comparison of viability in suspended and adherent cells (see next comment). As to architecture and other mechanistic analyses, these were beyond the scope of the new format.

Both adherent and suspended cells were used in the study. For adherent cells, did the authors confirm that transporter activity was equivalent in suspension versus surface-adhered conditions?

The trans-membranal transfer studies were all conducted in suspended cells. This was added to Methods (lines 115-116). The only analysis that was originally
conducted with adherent cells was the XTT assay. Based on this comment, we
compared the findings of this assays following a 2-hr exposure to microgravity or Earth conditions. No differences were identified between these treatment groups.

Reviewer 3 Report

Comments and Suggestions for Authors

The study presents the effect of microgravity on the diffusion of small molecules and effect on the different protein transporters membranes such as P-glycoprotein. The study presents the cellular effects of 7 small molecules known to have relevant effect on the transporters. The ex vivo study was conducted on 4 types of cell lines.

I do not recommend this paper to be published in Pharmaceutics, with a detection of a high similarity (almost 23%) by the Compilatio Software and for different major points, as follows:

1.     The manuscript presents a very brief introduction, that makes the citation of 21 references in 10 lines without further explanation.

2.     The studied transporters are not explained in the discussion, the conclusion in experimentations lacks detailed analysis.

3.     Many research papers are cited without a deep understanding of their choice and main input with the study.

4.     The choice of the components, their concentration and the control drugs is not presented.

5.      The manuscript is confusing, the main purpose of the study is presented in the 10th line but the conclusion cannot confirm the hypothesis.

6.     This topic clearly cannot sustain the novelty of the conclusion even though the authors tried to present the study’s limitations e.g. the duration of exposure.

Author Response

The study presents the effect of microgravity on the diffusion of small molecules and effect on the different protein transporters membranes such as P-glycoprotein. The study presents the cellular effects of 7 small molecules known to have relevant effect on the transporters. The ex vivo study was conducted on 4 types of cell lines. I do not recommend this paper to be published in Pharmaceutics, with a detection of a high similarity (almost 23%) by the Compilatio Software and for different major points, as follows:

The manuscript was mostly written from scratch. The only parts that could have resembled published text were those citing our own methodology and the section describing the RPM features based on the manufacture’s website, which was revised (lines 136-144).

1. The manuscript presents a very brief introduction, that makes the citation of 21 references in 10 lines without further explanation.

We added more background to the introduction and deleted several
references. Yet with the new format of a Brief Report, we preferred to
keep the Introduction as short as possible.

2. The studied transporters are not explained in the discussion, the conclusion in experimentations lacks detailed analysis.

We added referral to the studied transporters in the introduction (lines 50-51, 57-58). Table 1 (P. 2-3) provides a summary of the interaction of each probe with these transporters.

3. Many research papers are cited without a deep understanding of their choice and main input with the study.

The introduction was re-written.

4. The choice of the components, their concentration and the control drugs is not presented.

We added to Section 2.3 (lines 120-130) the rationale for each concentration. In several cases, e.g., Hoechst 33342, the literature presented a wide concentration range, and we chose the final concentration following several preliminary experiments.

5. The manuscript is confusing, the main purpose of the study is presented in the 10th line but the conclusion cannot confirm the hypothesis.

To improve clarity, we deleted the second part of the sentence that describes the hypothesis (lines 52-56). revised the entire Introduction, Discussion, and Conclusions sections.

6. This topic clearly cannot sustain the novelty of the conclusion even though the authors tried to present the study’s limitations e.g. the duration of exposure.

The novelty of this study is mostly in identifying several physicochemical properties that could affect the cellular uptake of smaller and larger molecules in SMG This was added to the Discussion (lines 309-311).

Round 2

Reviewer 1 Report

Comments and Suggestions for Authors

These are my comments:

1) Line 239: Issue with grammar “did not allow for achieving...” 2) Next line (line 240) as well, this is not a complete sentence, consider combining and re-writing so clear and correct grammatically. I believe they are trying to say that an exposure time of 1 hour did not allow them to reach the 0.001 simulated microgravity conditions, but that were able to reach 0.01 by 10 minutes and 0.006 g by 30 minutes. But then it says in the next sentence (line 241) that Lunar gravity was achieved within 30 min, which was 0.16 g, so its a bit confusing as this section is currently written. 3) Line 243 states that closed eppendorf tubes did not allow CO2 exchange (actually they say limited, but if they are closed as they are it would not allow any CO2 exchange, so this should be corrected). But then they note similar results were obtained with falcon flasks, however I am assuming these flasks would also have been closed, so would not allow CO2 exchange either. This needs to be clarified. 4) The conclusion on line 245 is not necessarily true, that the limitations would only result in UNDERESTIMATING the effect. It could also overestimate the effect, for example the bubbles causing shear forces, which are not typical under real microgravity may overestimate effects. This should also be noted. 5) Line 253 "Dimensionality should not…” is stated but this statement should be tempered to "Dimensionality may not be expected to affect our results, but these experiments should be performed to ensure this is the case.” Or something along these lines. Given what is being done in this prior experiment is not exactly the same as what is being done in the current work. 6) for Appendix E. Following the FSC/SSC gating the cells should be typically put through a doublet elimination strategy, was this performed? Please explain why “normalized to Mode” was used on the Y axis on (c) rather than the histogram (count).     Besides this I think most of the other comments were addressed. Comments on the Quality of English Language

These are my comments:

1) Line 239: Issue with grammar “did not allow for achieving...” 2) Next line (line 240) as well, this is not a complete sentence, consider combining and re-writing so clear and correct grammatically. I believe they are trying to say that an exposure time of 1 hour did not allow them to reach the 0.001 simulated microgravity conditions, but that were able to reach 0.01 by 10 minutes and 0.006 g by 30 minutes. But then it says in the next sentence (line 241) that Lunar gravity was achieved within 30 min, which was 0.16 g, so its a bit confusing as this section is currently written. 3) Line 243 states that closed eppendorf tubes did not allow CO2 exchange (actually they say limited, but if they are closed as they are it would not allow any CO2 exchange, so this should be corrected). But then they note similar results were obtained with falcon flasks, however I am assuming these flasks would also have been closed, so would not allow CO2 exchange either. This needs to be clarified. 4) The conclusion on line 245 is not necessarily true, that the limitations would only result in UNDERESTIMATING the effect. It could also overestimate the effect, for example the bubbles causing shear forces, which are not typical under real microgravity may overestimate effects. This should also be noted. 5) Line 253 "Dimensionality should not…” is stated but this statement should be tempered to "Dimensionality may not be expected to affect our results, but these experiments should be performed to ensure this is the case.” Or something along these lines. Given what is being done in this prior experiment is not exactly the same as what is being done in the current work. 6) for Appendix E. Following the FSC/SSC gating the cells should be typically put through a doublet elimination strategy, was this performed? Please explain why “normalized to Mode” was used on the Y axis on (c) rather than the histogram (count).     Besides this I think most of the other comments were addressed.

Author Response

Comment 1: Line 239: Issue with grammar “did not allow for achieving...”

Response: Corrected to “did not suffice“ (line 251).

Comment 2: Next line (line 240) as well, this is not a complete sentence, consider combining and re-writing so clear and correct grammatically. I believe they are trying to say that an exposure time of 1 hour did not allow them to reach the 0.001 simulated microgravity conditions, but that were able to reach 0.01 by 10 minutes and 0.006 g by 30 minutes. But then it says in the next sentence (line 241) that Lunar gravity was achieved within 30 min, which was 0.16 g, so its a bit confusing as this section is currently written.

Response: Corrected to “This issue was less pronounced with lunar gravity, because the target value (0.16 g) was achieved obtained within 30 minutes.“  (lines 253-255).

Comment 3: Line 243 states that closed eppendorf tubes did not allow CO2 exchange (actually they say limited, but if they are closed as they are it would not allow any CO2 exchange, so this should be corrected). But then they note similar results were obtained with falcon flasks, however I am assuming these flasks would also have been closed, so would not allow CO2 exchange either. This needs to be clarified.

Response: Thank you for the comment. The Falcon flasks were not tightly closed, and under this condition the allow gas exchange.  https://ecatalog.corning.com/life-sciences/b2b/CA/en/Cell-Culture/Cell-Culture-Vessels/Flasks%2C-Culture/Falcon%C2%AE-Cell-Culture-Flasks/p/falconCellCultureFlasks

This was also clarified in the text (lines 256-257).

Comment 4: The conclusion on line 245 is not necessarily true, that the limitations would only result in UNDERESTIMATING the effect. It could also overestimate the effect, for example the bubbles causing shear forces, which are not typical under real microgravity may overestimate effects. This should also be noted.

Response: Corrected to “misinterpret” (line 260).

Comment 5: Line 253 "Dimensionality should not…” is stated but this statement should be tempered to "Dimensionality may not be expected to affect our results, but these experiments should be performed to ensure this is the case.” Or something along these lines. Given what is being done in this prior experiment is not exactly the same as what is being done in the current work.

Response: Corrected to “Hence dimensionality is not expected to affect our results, but further experiments should be performed to ensure this is the case” (lines 268-270).

Comment 6: for Appendix E. Following the FSC/SSC gating the cells should be typically put through a doublet elimination strategy, was this performed? Please explain why “normalized to Mode” was used on the Y axis on (c) rather than the histogram (count).

Response: The strategy was not used because cells were incubated for one hour only, such that cell cycle-related analyses were less relevant. We used the “Normalization to Mode” strategy to allow easier appreciation of shifts to the left or the right (lines 183-184).

Reviewer 2 Report

Comments and Suggestions for Authors

The authors' responses to my previous comments are unsatisfactory. They lack careful consideration, and some of the comments were not addressed at all. I do not believe this manuscript is suitable for publication. Below are a few specific concerns:

 1.      The authors are unwilling to perform any mechanistic studies and have reported cellular uptake in different cell lines that are less pharmacologically relevant for drug absorption.

2.      Authors did not respond properly to my previous comment- For adherent cells, did the authors confirm that transporters activity was equivalent in suspension versus surface-adhered conditions?

3.      Authors did not respond to previous concern- In section 3.3, cellular uptake was studied with only one molecule, and the confocal images have different backgrounds for the red color. These images need to be captured under higher power for accurate calculation, and a single molecular evaluation is not sufficient to draw conclusions about the class of drugs showing transporter-mediated uptake. Additionally, the Western blots in Figure 3d are not labeled for the proteins.

4.      The revised manuscript contains numerous typographical errors, and the writing still needs improvement. Additionally, the figure captions for Figures 2 and 3 have been deleted without new captions being added. The reference numbers are not in the correct order.

Comments on the Quality of English Language

The revised manuscript contains numerous typographical errors, and the writing still needs improvement.

Author Response

Comment 1: The authors are unwilling to perform any mechanistic studies and have reported cellular uptake in different cell lines that are less pharmacologically relevant for drug absorption.

Response: In line with the Scientific Editor suggestion, the manuscript format was modified to a Brief Report that presents a pilot study. Therefore, mechanistic studies were beyond the scope of this work.

As to the relevance of the cell lines, the focus was not necessarily absorption. This was stated in the hypothesis, as follows: “we hypothesized that microgravity could affect the distribution of small molecules into cells” (lines 44-45). The rationale for cell selection is in line with this hypothesis, and is explained in Section 2.2 (lines 82-88). We also added a referral to other pharmacokinetic changes in the discussion (lines 222-223).

Comment 2: Authors did not respond properly to my previous comment- For adherent cells, did the authors confirm that transporters activity was equivalent in suspension versus surface-adhered conditions?

Response: As we explained in the previous version, all the studies other than the original viability study were conducted in suspended cells (lines 98-99). The comparison to adherent cells was beyond the scope of the current study and will be conducted in the future.

Comment 3: Authors did not respond to previous concern- In section 3.3, cellular uptake was studied with only one molecule, and the confocal images have different backgrounds for the red color. These images need to be captured under higher power for accurate calculation, and a single molecular evaluation is not sufficient to draw conclusions about the class of drugs showing transporter-mediated uptake. Additionally, the Western blots in Figure 3d are not labeled for the proteins.

Response: The only conclusion we made regarding GB123 was “The SMG effect on a larger molecule, GB123, was similar to those observed with the efflux transporter substrates (although GB123 is not a P-gp or BCRP substrate)” (lines 243-244). This does not make any generalizations.

We replaced the images with ones that were obtained at higher magnification. The image settings were added, to demonstrate that images represent the same thresholds. Image analysis is independent of these definitions that affect visuality only. As to the apparent differences in the red background, this could be due to cathepsin leakage from cells (Hamed et al., 2024; PMID 38371846). The proteins are labeled in the figure as “CTS B”, “CTS S”, and “CTS L” to the left side of the bands in (d), with a related legend (lines 349-350). 

Comment 4: The revised manuscript contains numerous typographical errors, and the writing still needs improvement. Additionally, the figure captions for Figures 2 and 3 have been deleted without new captions being added. The reference numbers are not in the correct order.

Response: The manuscript was re-edited by a native speaker of English.

The captions were above the figures to reflect the Appendix format. However, to improve clarity, they are now below the figures.

The references were re-reviewed. In some cases, the reference is cited more than once and numbering might therefore appear incorrect.